# MEG-GPT-2: A Generative Foundation Model for MEG with Structured Neural Representations

**SungJun Cho** [1] [2]  **Chetan Gohil** [1]  **Oiwi Parker Jones** [2]  **Mark Woolrich** [1]

## Abstract

Foundation modelling of neural time series has largely focused on EEG and is typically evaluated with downstream tasks, leaving open questions about the fidelity of learnt representations. In this paper, we introduce MEG-GPT-2, a decoder-only transformer-based foundation model trained on source-level MEG data using a self-supervised autoregressive objective with factorised channel-time attention. Post-hoc analyses show that the model successfully captures temporal dynamics and spectral structure of brain activity while encoding anatomically meaningful spatial organisation and partially recovering amplitude-based functional connectivity. However, phase-sensitive inter-channel dependencies remain insufficiently modelled, highlighting limitations in capturing higher-order dynamics under short context lengths. Our results demonstrate that generative and representation-level evaluations provide complementary insights beyond downstream performance, offering a framework for assessing the neuroscientific validity of foundation models.

## 1. Introduction

Foundation modelling of neural time series has emerged as a key challenge in neural decoding and brain–computer interfaces. A central objective of these models is to learn generalisable representations of neural activity that transfer across tasks with applications in both clinical settings and fundamental neuroscience. However, despite recent progress in foundation models for non-invasive human electrophysiology data, key limitations remain.

First, most existing approaches focus on electroencephalography (EEG), mainly due to the availability of large-scale datasets. In contrast, magnetoencephalography (MEG) remains underexplored due to limited data availability and higher acquisition cost. Only a small number of MEG-specific models have been proposed, and MEG is often incorporated alongside EEG in multimodal frameworks[1]. Yet, MEG provides a higher signal-to-noise ratio and spatial resolution than EEG (Gross, 2019; Cho et al., 2024), making it a promising modality for foundation modelling, particularly for learning robust representations across data distributions.

Secondly, existing models are primarily evaluated on downstream task performance, which may not directly reflect the quality of learnt representations. High task performance does not necessarily imply that models capture intrinsic properties of neural signals, and task-specific optimisation during model design can introduce inductive biases that limit generalisation. Consequently, there is a need for evaluation paradigms that assess whether models are capturing fundamental neural characteristics. To our knowledge, there is limited evidence demonstrating that current models capture intrinsic neural characteristics such as spectral power, phase structure, or functional and structural connectivity.

To address these limitations, we propose MEG-GPT-2, a decoder-only transformer foundation model for MEG time series. The model incorporates sequential channel and temporal attention, trained using a self-supervised autoregressive next-token prediction objective. We investigate whether a generative modelling approach of this form can capture statistical and dynamical structure in neural signals, including power, phase, and connectivity patterns. Our results suggest that it successfully captures spectral and temporal dynamics while partially recovering functional connectivity.

## 2. Methodology

### 2.1. Data Tokenisation

To tokenise data, we adopt the learnable, fully causal tokeniser introduced in Cho et al. (2026), which operates at the

---

[1]Oxford Centre for Human Brain Activity, Department of Psychiatry, University of Oxford [2]PNPL, Oxford Robotics Institute, University of Oxford. Correspondence to: SungJun Cho <sungjun.cho@ndcn.ox.ac.uk>.

*Proceedings of the 2nd ICML Workshop on Foundation Models for Structured Data*, Seoul, South Korea. 2026. Copyright 2026 by the author(s).

[1]See Appendix A for a detailed discussion of related work and references.

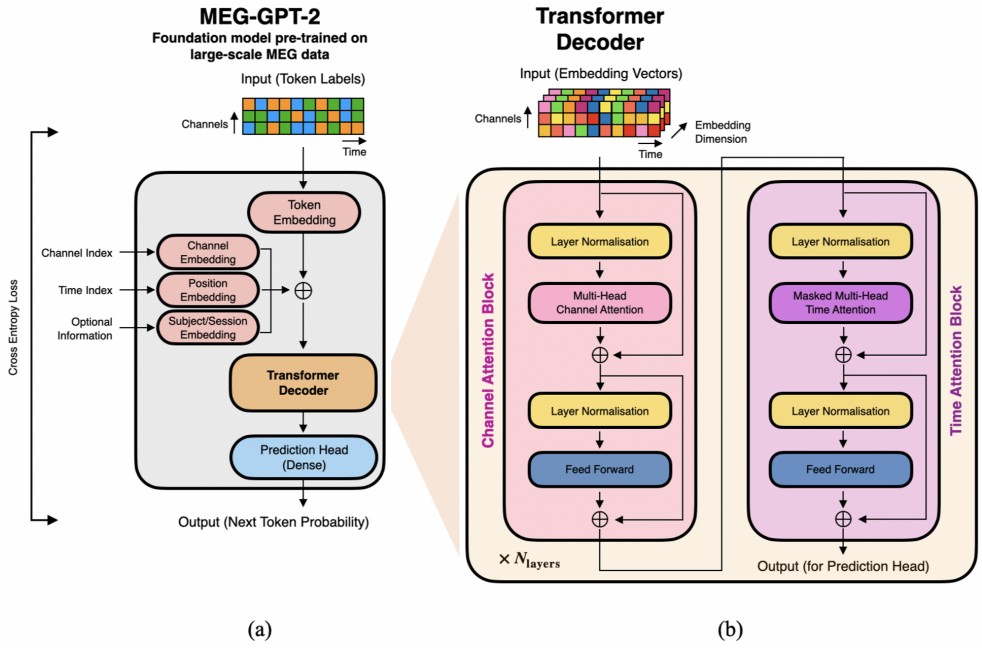

*Figure 1.* MEG-GPT-2 architecture. (a) Overall model pipeline. Discrete token sequences are mapped to embeddings and augmented with channel, positional, and optional subject/session embeddings. The resulting representations are processed by a decoder-only transformer, followed by a dense prediction head that outputs next-token probabilities under an autoregressive objective. (b) Transformer decoder block. Each layer factorises attention into two sequential sub-blocks: (i) multi-head *channel attention*, modelling inter-channel dependencies at each time step, and (ii) masked multi-head *temporal attention*, enforcing causal dependencies along the time dimension. Both sub-blocks adopt a pre-layer normalisation design and consist of self-attention and position-wise feedforward networks with residual connections (denoted by $\oplus$). The block is repeated for $N_{\text{layers}}$ layers.

sample level (i.e., one token per time step) and is designed to preserve temporal dependencies and spectral characteristics of M/EEG signals. The tokeniser maps continuous-valued inputs to discrete token representations in a causal manner, aligning with the autoregressive training objective of MEG-GPT-2. Full implementation details and hyperparameters are provided in Cho et al. (2026).

## 2.2. Model Network Architecture

### 2.2.1. INPUT EMBEDDINGS

Each input token is represented as the sum of multiple learnt embeddings capturing complementary structure: token identity, temporal position, channel (brain parcel), and optional subject/session metadata. For channel $c \in \{1, \ldots, C\}$ and time index $t \in \{1, \ldots, L\}$, embeddings are obtained via lookup tables:

- Token embedding: $e_{\text{token}} \in \mathbb{R}^{K \times d_{\text{token}}}$
- Positional embedding: $e_{\text{pos}} \in \mathbb{R}^{L \times d_{\text{pos}}}$
- Channel embedding: $e_{\text{ch}} \in \mathbb{R}^{C \times d_{\text{ch}}}$
- Subject embedding: $e_{\text{subj}} \in \mathbb{R}^{N \times d_{\text{subj}}}$

where $K$ is the token vocabulary size and $N$ is the number of subjects. All embeddings are projected to the model dimension $d_{\text{model}}$ when necessary via linear projections. The final input representation is given by the sum of all embeddings, which is then passed to the transformer (Figure 1a).

### 2.2.2. MODEL ARCHITECTURE

MEG-GPT-2 is a decoder-only transformer composed of $N_{\text{layers}}$ stacked layers (Figure 1b). Each layer implements a factorised self-attention mechanism over channel and temporal dimensions. Given hidden states $\mathbf{h}^l \in \mathbb{R}^{C \times L \times d}$, the layer applies:

$$\tilde{\mathbf{h}} = \text{Attn}_c(\text{LN}(\mathbf{h}^l)) + \mathbf{h}^l, \tag{1}$$

$$\check{\mathbf{h}} = \text{FFN}_c(\text{LN}(\tilde{\mathbf{h}})) + \tilde{\mathbf{h}}, \tag{2}$$

$$\hat{\mathbf{h}} = \text{Attn}_t(\text{LN}(\check{\mathbf{h}})) + \check{\mathbf{h}}, \tag{3}$$

$$\mathbf{h}^{l+1} = \text{FFN}_t(\text{LN}(\hat{\mathbf{h}})) + \hat{\mathbf{h}}, \tag{4}$$

where $\text{Attn}_c$ denotes multi-head channel attention (across channels at fixed time) and $\text{Attn}_t$ denotes masked multi-

head temporal attention (across time within each channel). Here, temporal attention is strictly causal.

Channel and temporal attention are applied sequentially, enabling structured information mixing: inter-channel dependencies are first aggregated at each time step, followed by temporal propagation within each channel. All sub-blocks adopt a pre-layer normalisation design with residual connections.

To improve computational efficiency and encourage hierarchical temporal modelling, temporal attention employs (i) patching, (ii) sparse banded attention over keys (Child et al., 2019), and (iii) Perceiver AR-style query truncation (Hawthorne et al., 2022). This progressively expands the receptive field across layers, allowing early layers to focus on local dynamics and deeper layers to capture long-range dependencies. Illustrations of the attention masks are provided in Appendix B.

Dropout is optionally applied after feedforward layers along channel-wise and element-wise dimensions for regularisation.

### 2.2.3. LEARNING OBJECTIVES

MEG-GPT-2 is trained using an autoregressive next-token prediction objective. Given predicted probabilities $\hat{p}_{t,c,k}$ over the token vocabulary $k \in \{1, \ldots, K\}$, the loss is defined as the average cross-entropy over channels and a subset of time steps:

$$\mathcal{L} = -\frac{1}{|\mathcal{T}|\, C} \sum_{t \in \mathcal{T}} \sum_{c=1}^{C} \sum_{k=1}^{K} \mathbf{1}[y_{t,c} = k] \log \hat{p}_{t,c,k}, \quad (5)$$

where $\mathcal{T} = \{L - L_{\text{loss}} + 1, \ldots, L\}$.

We restrict the loss to the final $L_{\text{loss}}$ time steps to mitigate instability from early predictions, which are conditioned on shorter contexts. This improves optimisation stability and better aligns training with the intended longer-context autoregressive regime.

### 2.3. Synthetic Data Generation

Synthetic MEG time series are generated using a pre-trained MEG-GPT-2 model, conditioned on data from the first 50 subjects. For each sequence, an initial prompt is constructed from the corresponding tokenised data using a window of length equal to the model context size. Generation proceeds autoregressively using nucleus (top-$p$) sampling with $p = 0.9$ and temperature $\tau = 1.5$. At each step, a token is sampled and the input window is shifted forward to maintain a fixed-length context. This process is repeated to produce full-length sequences, which are then decoded via the tokeniser to obtain continuous MEG signals.

## 3. Experiments & Results

### 3.1. Dataset & Data Preparation

We pre-trained our model on the Cambridge Centre for Ageing and Neuroscience (Cam-CAN) dataset (Shafto et al., 2014), which comprises resting-state, eyes-closed MEG recordings from 612 healthy participants. A subset of 50 subjects was randomly subsampled to train the tokeniser, which is subsequently applied to the full data to generate tokenised sequences for training MEG-GPT-2.

Importantly, we operate on source-level rather than sensor-level MEG data. The recordings were source-reconstructed and parcellated into anatomically meaningful brain regions, such that each channel corresponds to a specific cortical parcel. Data at a source level provides biologically interpretable structure and facilitates analysis of learnt representations in relation to anatomical regions. Further details on demographics, preprocessing, and source reconstruction are provided in Appendix C.

### 3.2. Model Training

MEG-GPT-2 was implemented in Python 3.10 using the PyTorch library (v2.5.1) and trained on two NVIDIA A100 GPUs, requiring approximately 47 GPU hours. Model performance and parameter counts are reported in Table 1, alongside comparisons with MEG-GPT (Huang et al., 2025). Further descriptions on the training procedure and a complete list of hyperparameters is provided in Appendix D.2.

### 3.3. Interpretability Analysis of Spatial Information

We examined how spatial structure is represented in MEG-GPT-2 through two components: channel embeddings and channel attention matrices. As the model is not explicitly constrained to encode anatomical or functional structure, this analysis provides a post-hoc assessment of emergent representations.

We first computed pairwise cosine similarity between channel embedding vectors, yielding a matrix in $\mathbb{R}^{C \times C}$ (Figure 2a). The resulting structure exhibited clear block patterns corresponding to cortical subdivisions. This indicates that the embeddings capture geometric relationships induced by the underlying parcellation, implying that the model internalises spatial priors without explicit supervision.

To probe functional interactions, we then extracted channel attention values by performing a forward pass on the validation set and averaging attention matrices across batch, time steps, and attention heads. A representative matrix from Layer 5 is shown in Figure 2b, with results across all layers provided in Appendix E.1. Compared to the embedding similarity, channel attention exhibits more diffuse but structured patterns, with localised clusters of elevated weights

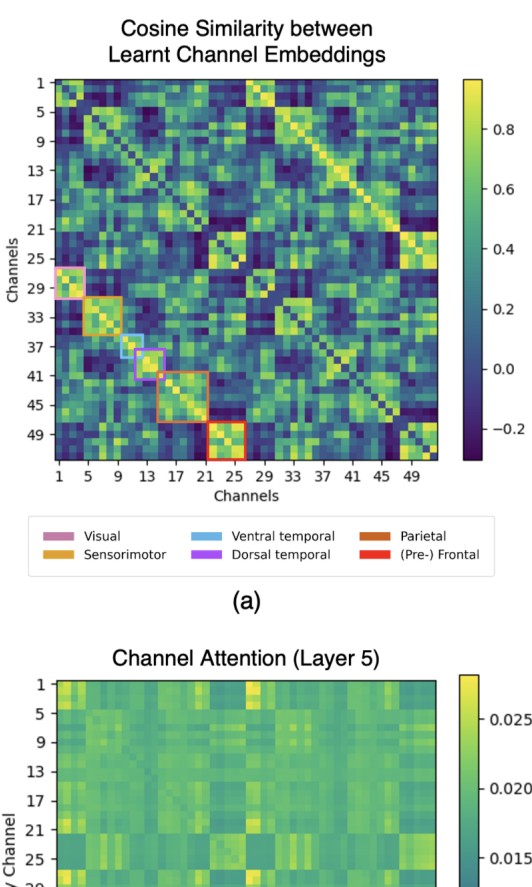

(a)

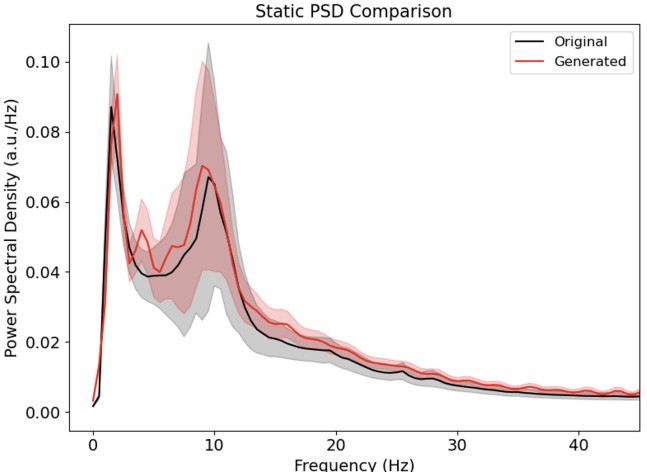

(b)

*Figure 2.* Post-hoc analysis of spatial representations. (a) Cosine similarity matrix of learnt channel embeddings. Block structures reflect grouping of anatomically related cortical regions, with symmetric patterns indicating bilateral correspondence across hemisphere. (b) Channel attention matrix (Layer 5), averaged over batch, time, and attention heads on the validation set.

across subsets of channels. These patterns suggest that the model captures inter-channel dependencies that may reflect functional connectivity.

### 3.4. Post-hoc Analysis of Generated Synthetic Data

Next, we evaluated whether synthetic data generated by the pre-trained model preserves key spectral and spatiotemporal properties of the original MEG signals. Figure 3 shows group-averaged power spectral densities (PSDs) for real and

generated data. The generated signals closely match the overall spectral profile of the real data, including the characteristic low-frequency dominance and a prominent peak in the alpha band. This indicates that the model captures global spectral structure and temporal dynamics effectively. Additional qualitative comparisons are provided in Appendix F.

*Figure 3.* Power spectral density (PSD) comparison between real and generated data. PSDs are averaged over channels and subjects. Solid lines denote the mean, and shaded regions indicate standard deviation.

We further assessed whether inter-channel dependencies are preserved in the generated data. Using amplitude envelope correlation as a proxy for functional connectivity, we observed partial reconstruction of connectivity structure (Appendix E.2), although the patterns were less pronounced compared to real data. In contrast, analyses based on time-delay embedded covariance failed to recover consistent structure, suggesting that phase relationships and higher-order temporal dependencies are not well captured.

## 4. Conclusion

In this paper, we presented MEG-GPT-2, a decoder-only transformer foundation model for MEG time series. Our analyses show that it captures spectral and temporal dynamics of brain activity while encoding anatomically meaningful spatial structure and some amplitude-based functional connectivity. However, phase-sensitive inter-channel dependencies remain insufficiently modelled. Our results further highlight the value of generative and representation-level evaluations, which provide complementary insights beyond downstream task performance. Future work will focus on scaling model capacity and context length to better capture long-range dependencies and functional connectivity, as well as benchmarking on downstream tasks.

## Software and Data

The Cam-CAN dataset is publicly available at Shafto et al. (2014) and Taylor et al. (2017). Detailed descriptions of the dataset and acquisition protocol can be found in these papers. Data collection was conducted in accordance with the Declaration of Helsinki and approved by the Cambridgeshire 2 Research Ethics Committee. All participants provided written informed consent.

The code for the tokeniser and MEG-GPT-2 is available in https://github.com/OHBA-analysis/EphysTokenizer and https://github.com/OHBA-analysis/MEG-GPT, respectively.

## Acknowledgements

We thank the reviewers for their helpful comments and suggestions. This work was supported by the Medical Research Council (MR/W006731/1, MR/X00757X/1, and the New Therapeutics in Alzheimer's Disease (NTAD) programme), the Wellcome Trust (106183/Z/14/Z and 215573/Z/19/Z), Dementias Platform UK (RG94383 and RG89702), the Royal Society (RG/R1/241267), the National Science Foundation (2314493), the NFRF (NFRFT-2022-00241), the SSHRC (895-2023-1022), the NIHR Oxford Health Biomedical Research Centre (NIHR203316), the Advanced Research + Invention Agency (SCNI-SE01-P004), the Hertford Claire Clifford Lusardi Scholarship, and the Nuffield Department of Clinical Neurosciences. The views expressed are those of the authors and do not necessarily reflect those of the NIHR or the UK Department of Health and Social Care.

## Impact Statement

This paper presents work whose goal is to advance the field of machine learning, neuroscience, and brain-computer interfaces. There are many potential societal consequences of our work, none of which we feel must be specifically highlighted here.

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

## A. Related Works

Recent years have witnessed rapid growth in foundation models for human brain electrophysiology data. Most existing approaches are built on the transformer architecture and focus primarily on EEG. Broadly, these models can be categorised into three classes based on the model architecture (Cho et al., 2026):

- **Encoder-only** transformers trained to predict masked tokens (e.g., LaBraM (Jiang et al., 2024); CBraMod (Wang et al., 2025); BIOT (Yang et al., 2023))

- **Encoder-decoder** masked autoencoders trained for signal reconstruction (e.g., REVE (Ouahidi et al., 2025))

- **Decoder-only** autoregressive transformers trained with next-token prediction objectives (e.g., Neuro-GPT (Cui et al., 2024))

While recent work on EEG has also explored some feature attribution methods, these analyses are typically conditioned on task labels and remain task-specific (Wang et al., 2025; Xiong et al., 2025; Guo et al., 2026).

In contrast, foundation modelling for MEG stands relatively underexplored. Only a limited number of MEG-specific models such as BBL (Jayalath et al., 2025), MEG-XL (Jayalath & Parker Jones, 2026), and MEG-GPT (Huang et al., 2025) have been proposed, and multimodal approaches integrating EEG and MEG are similarly scarce (cf. Brain-Omni (Xiao et al., 2025); Brain-OF (Guo et al., 2026)). Comprehensive surveys and benchmarks, though largely on EEG, are available in Kuruppu et al. (2025), Wu et al. (2025), Xiong et al. (2025), and Zhou et al. (2025), with Liu et al. (2026) reporting over 50 EEG foundation models, providing a useful overview of the current landscape.

# B. Channel and Time Attention Masks

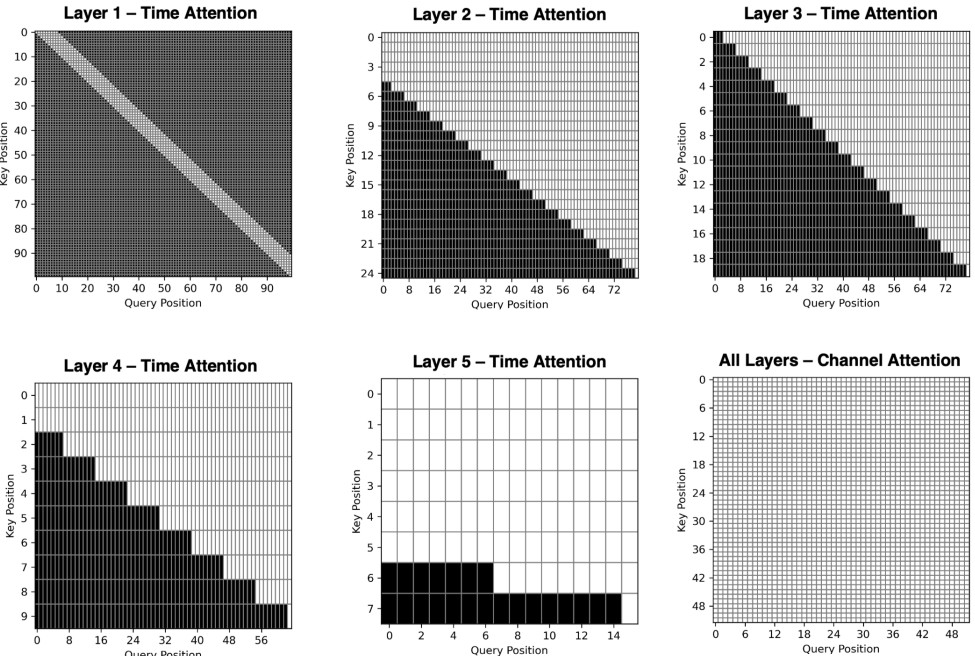

*Figure 4.* Attention masks used in MEG-GPT-2. Top and bottom panels (left to right) show temporal attention masks for the transformer layers 1–5. Temporal attention is strictly causal and employs a progressively expanding receptive field: early layers attend only to a narrow local context (banded structure near the diagonal), while deeper layers access increasingly longer temporal dependencies via patching and query truncation. This hierarchical design enables the model to capture short-range dynamics in lower layers and long-range dependencies in higher layers within a fixed context length. Rightmost panel shows the channel attention mask, which is dense and identical across all layers, allowing full interaction across channels at each time step. Black denotes masked positions, while white denotes unmasked positions. Note that the query and key indices at the axes are simply illustrative and do not correspond to absolute temporal positions. The Perceiver AR-style truncation is applied to the query vectors, while patching and sparse banding are applied to the key vectors.

## C. Dataset Details

The Cam-CAN dataset (Shafto et al., 2014) contains resting-state, eyes-closed MEG data collected from 612 healthy participants (aged 18-88 years; 310 males, 302 females). Each scan lasted approximately 8.5 minutes and was measured using a 306-channel Elekta scanner at a sampling frequency of 1 kHz.

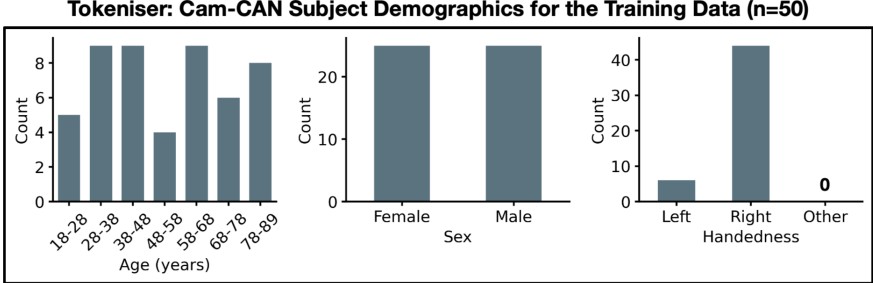

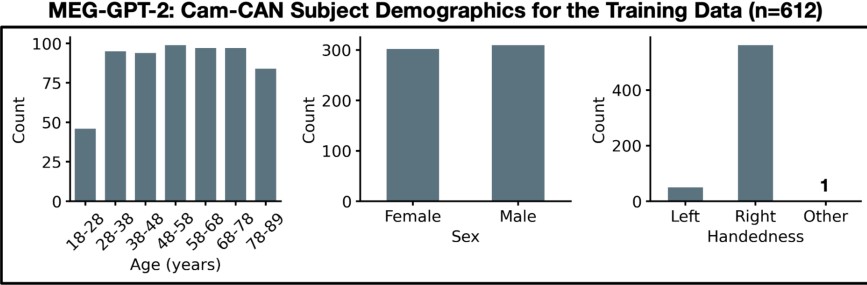

*Figure 5.* Subject demographics of the training data for the tokeniser and MEG-GPT-2.

### C.1. Preprocessing

All datasets were preprocessed and source-reconstructed using the *osl-ephys* toolbox (van Es et al., 2025). Raw MEG recordings were first band-pass filtered between 0.5-125 Hz and notch-filtered to remove power-line interference at 50 and 100 Hz, followed by downsampling to 250 Hz. Noisy channels and time segments were automatically identified and excluded using the generalised extreme Studentised deviate (GESD) algorithm (Rosner, 1983).

We then applied independent component analysis (ICA) to the data using FastICA (Hyvarinen, 1999), decomposing the sensor-level data to 64 components. Components exhibiting high correlation with electrooculogram (EOG) or electrocardiogram (ECG) signals were classified as artefacts and removed.

### C.2. Source Reconstruction

Preprocessed sensor-level data were co-registered and projected to source space on an 8-mm isotropic grid using a volumetric linearly constrained minimum variance (LCMV) beamformer (Veen & Buckley, 1988). The resulting voxel-wise time series were parcellated into 52 anatomically defined regions based on the Glasser52 atlas (Kohl et al., 2023). To mitigate source leakage and spurious inter-parcel correlations, symmetric multivariate leakage reduction (Colclough et al., 2015) was applied, removing zero-lag correlations between parcel time series. Each parcel time series was subsequently standardised (zero mean, unit variance) along the temporal dimension. All model training and analyses were conducted on these standardised source-level signals.

# D. Model Training Details

## D.1. Pre-training Results

Table 1 reports pre-training performance of MEG-GPT-2 compared to MEG-GPT. Due to computational constraints, results are currently based on a single training run per model. Both models are trained on the same dataset using an identical tokeniser, ensuring a controlled comparison.

MEG-GPT-2 achieves lower training and validation loss, as well as higher top-1 accuracy, despite having a comparable number of parameters. While these results suggest improved or comparable learning efficiency, we note that conclusions are preliminary given the limited number of runs.

*Table 1.* Model pre-training performance metrics
(Abbreviation – acc.: accuracy, params.: number of parameters)

| METRIC | MEG-GPT-2 | MEG-GPT |
|---|---|---|
| TRAIN LOSS | 1.710 | 1.719 |
| TRAIN TOP-1 ACC. | 0.340 | 0.339 |
| VALIDATION LOSS | 1.700 | 1.721 |
| VALIDATION TOP-1 ACC. | 0.343 | 0.338 |
| PARAMS. | 7.3M | 7.4M |

## D.2. Model Hyperparameters

The hyperparameters used for training and evaluation (Section 3) are summarised in Table 2. Due to computational and memory constraints, we employed a context length of 100 tokens.

We hypothesise that scaling both model capacity and context length would improve the model's ability to capture long-range dependencies, particularly for learning functional connectivity in neural signals. Prior work on MEG foundation modelling has demonstrated the benefits of longer contexts in downstream tasks such as speech decoding (Jayalath & Parker Jones, 2026). Combined with increased dataset scale, we expect further gains in representation quality and overall model performance.

**Training Procedure**  For each subject in the Cam-CAN dataset, tokenised data were split into training and validation sets with a 9:1 ratio. The model was trained using the AdamW optimiser (Loshchilov & Hutter, 2019) with a learning rate of 5e-5 and momentum parameters $\beta_1 = 0.9$ and $\beta_2 = 0.98$. A cosine annealing schedule with linear warm-up was applied to the learning rate. Training was conducted with a batch size of 128 and a context length of 100 tokens. Input embeddings had dimensionality of 256. The model architecture comprised five transformer decoder layers, each with four self-attention heads for channel or time attention. The prediction head consisted of a feedforward network with 1024 hidden units.

*Table 2.* MEG-GPT-2 Model and Training Configuration

| Hyperparameter | Value |
| --- | --- |
| ***Base Configuration*** | |
| Sequence Length | 100 |
| Number of Channels | 52 |
| ***Input Embedding*** | |
| Number of Tokens | 89 |
| Embedding Dimension | 256 |
| Token / Positional / Channel Embedding Dim. | 256 / 256 / 256 |
| Positional Embedding Type | Absolute (Learnable) |
| Subject Embedding Dimension | 256 |
| ***Transformer Decoder Architecture*** | |
| Number of Heads | 4 |
| Model Dimension | 256 |
| Feed-Forward Dimension / Activation | 1024 / GELU |
| Normalisation Type | Layer |
| Dropout | 0.2 |
| Number of Input Patches | $[100, 25, 20, 10, 8]$ |
| Input Patch Lengths | $[1, 4, 4, 8, 8]$ |
| Number of Output Patches | $[25, 20, 10, 8, 16]$ |
| Output Patch Lengths | $[4, 4, 8, 8, 1]$ |
| Per-Layer Channel Attention | [False, True, True, True, True] |
| Per-Layer Sparse Banding Length | $[8, 0, 0, 0, 0]$ |
| Full Channel Attention Dropout | 0.1 |
| Channel Attention Channel Dropout | 0.1 |
| ***Loss & Objectives*** | |
| Loss Sequence Length | 16 |
| ***Training & Optimisation*** | |
| Batch Size | 128 |
| Number of Epochs | 40 |
| Train/Validation Split | 9:1 |
| Optimiser | AdamW |
| Learning Rate (LR) | $5.0 \times 10^{-5}$ |
| Weight Decay | 0.01 |
| Adam Betas $(\beta_1, \beta_2)$ / Epsilon $(\epsilon)$ | $[0.9, 0.98]$ / $1.0 \times 10^{-8}$ |
| Gradient Norm Clipping | 1.0 |
| LR Scheduler | Cosine Annealing w/ Linear Warmup |
| Total Steps / Warmup Steps | 224,200 / 22,420 |
| Warmup Start LR / Min LR $(\eta_{\min})$ | $1.0 \times 10^{-7}$ / $1.0 \times 10^{-6}$ |

# E. Analysis of Spatial Information in MEG-GPT-2

## E.1. Channel Attention

Figure 6 shows channel attention matrices extracted from the validation set across different transformer layers. Attention values are averaged over batch, time steps, and attention heads to provide a global view of inter-channel interaction patterns.

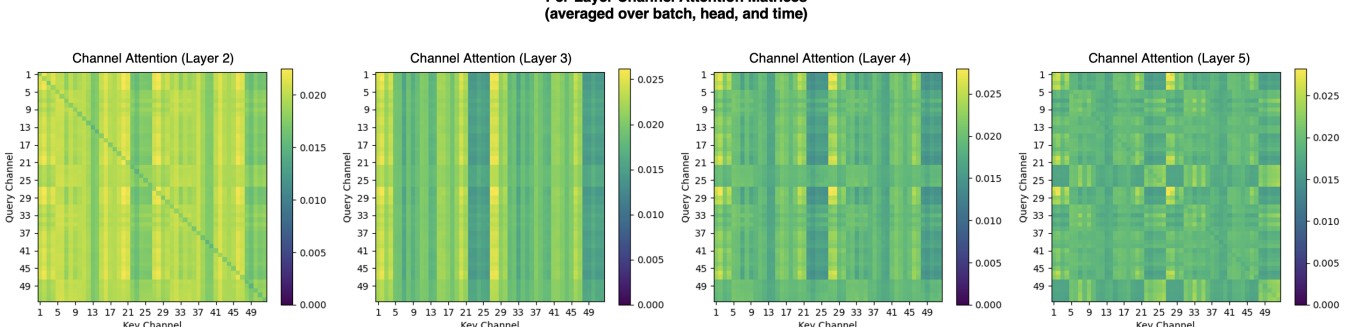

*Figure 6.* Per-layer channel attention matrices. Attention values are averaged over batch, time, and attention heads on the validation set. No channel attention is applied in Layer 1 by design (cf. Table 2), as the first layer focuses on modelling temporal dynamics prior to inter-channel interactions.

In intermediate layers (Layers 2–3), attention distributions appear relatively uniform across query channels, indicating limited specialisation and suggesting that channel-wise interactions are not yet strongly differentiated. In contrast, deeper layers (e.g., Layer 5) exhibit more structured patterns, with localised clusters of higher attention values across subsets of channels. This progression is consistent with a hierarchical representation, where deeper layers capture more complex and potentially functionally relevant inter-channel dependencies.

We note that this analysis is based on averaged attention maps; finer-grained structure may be present at the level of individual heads or time steps. A more detailed analysis of head-wise attention patterns and their temporal dynamics could provide additional insights, and we plan to address this in our future work.

## E.2. Functional Connectivity

As discussed in Section 3.4, we evaluated whether MEG-GPT-2 preserves inter-channel dependencies by comparing functional connectivity (FC) measures computed from real and generated data. Specifically, we consider amplitude envelope correlation (AEC) and time-delay embedded (TDE) covariance, which capture amplitude-based and phase-sensitive interactions, respectively.

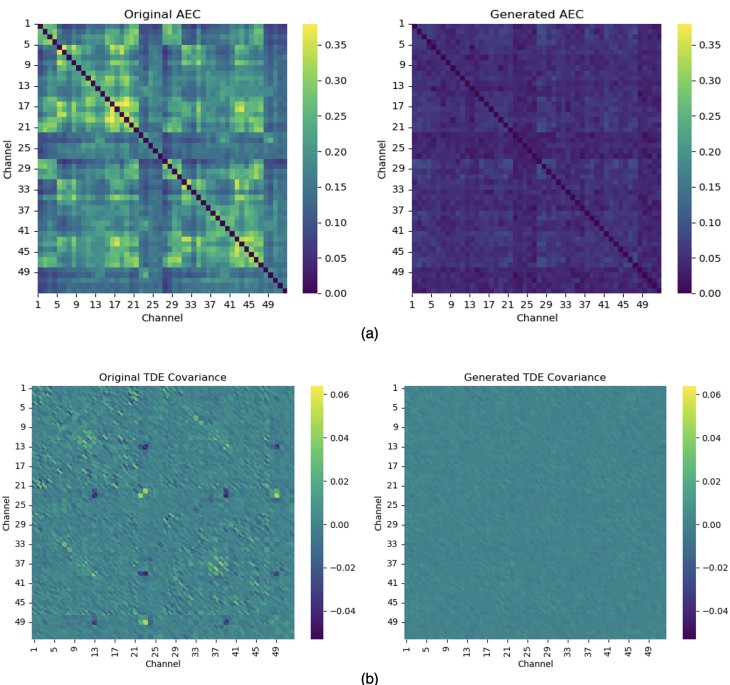

*Figure 7.* Functional connectivity comparison between real and generated data. (a) Amplitude envelope correlation (AEC) computed from band-limited signals (1–45 Hz). (b) Time-delay embedded (TDE) covariance matrices (number of embeddings=15) capturing spatio-temporal dependencies and dynamical structures within the time series data.

Figure 7a shows AEC matrices computed from band-limited signals (1–45 Hz). The generated data exhibits partial reconstruction of the connectivity structure observed in the real data, with spatially coherent patterns across nearby cortical regions. This suggests that the model captures amplitude-dependent correlations, although weak, which are often associated with large-scale co-modulation of neural activity. Notably, AEC patterns are qualitatively similar to the embedding-based cosine similarity (Figure 2a), reflecting the tendency of anatomically proximal regions exhibiting correlated amplitude fluctuations.

In contrast, TDE covariance matrices (Figure 7b) show limited correspondence between real and generated data. As TDE covariance incorporates temporal lags and phase relationships, this result indicates that the model does not adequately capture phase-dependent or higher-order temporal interactions across channels.

Along with Section 3.4, these results indicate that while the model successfully reproduces first-order temporal statistics and spectral characteristics, it remains limited in modelling cross-channel interactions. Although MEG-GPT-2 could model amplitude-driven functional connectivity to some extent, it remains limited in representing phase-sensitive inter-channel dynamics. This discrepancy could be possibly influenced by the relatively short context length (100 tokens, ∼0.4 s), which constrains the ability to model long-range dependencies. Nevertheless, the findings highlight that generative evaluation (beyond downstream task performance) provides a complementary perspective on whether learnt representations encode neuroscientifically meaningful structure.

## F. Analysis of Synthetic Signals Generated by MEG-GPT-2

Figure 8 presents a qualitative comparison between real and generated signals for a representative subject and channel (i.e., right temporal cortex), including time series, time–frequency spectrograms, and PSDs.

The generated time series exhibits realistic temporal variability, with amplitude fluctuations comparable to the original signal. In the time–frequency domain, both real and generated spectrograms show dominant low-frequency activity with intermittent burst-like patterns, indicating that the model captures non-stationary temporal dynamics.

The PSD further confirms spectral fidelity, with the generated signal reproducing the characteristic low-frequency dominance and peak structure observed in the real data, consistent with the group-level results in Figure 3. Notably, the model also reproduces the attenuation around 50 Hz, reflecting the notch filtering of power-line interference applied during preprocessing.

Overall, these results suggest that MEG-GPT-2 can generate signals with realistic temporal and spectral properties at the single-channel level.

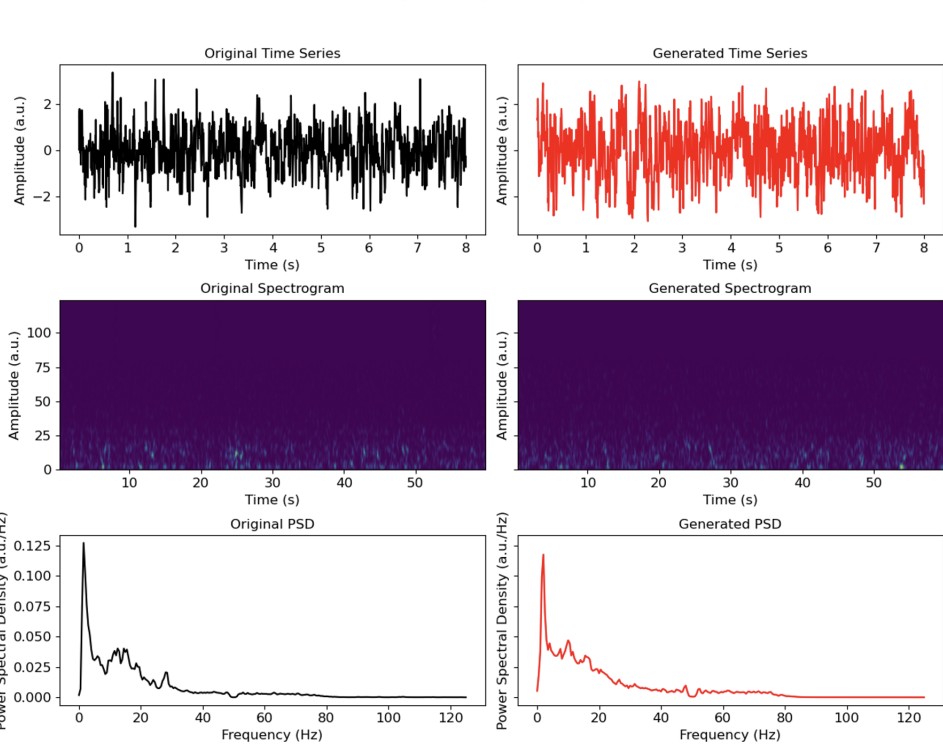

*Figure 8.* Single-channel comparison of real (left) and generated (right) signals. Time series, time–frequency spectrogram, and power spectral density (PSD) for a representative subject and channel (right temporal cortex).

