# OpenReview forum: "MEG-GPT-2: A Generative Foundation Model for MEG with Structured Neural Representations"
_ICML.cc/2026/Workshop/FMSD — FMSD @ ICML 2026 Poster_

### Official Review · Reviewer_vBDf · 2026-05-18
**Submission 123 Review**

**Rating:** 4
**Confidence:** 5

**Review:**

## Summary

The paper proposes MEG-GPT2, an autoregressive decoder-based model for MEG signals. The authors use the model to generate synthetic MEG data and compare it qualitatively with original signals using characteristics such as functional connectivity, spectral density, and related neurophysiological properties.

## Strengths

- The paper proposes an interesting approach for generating synthetic data for MEG signals. Since MEG data is difficult and expensive to obtain at scale, the proposed approach could serve as a promising data augmentation and synthetic data generation alternative. Such approaches could potentially support the training of large-scale MEG foundation models or complement existing EEG-based foundation models such as LaBraM and NeuroLM.

- The qualitative post-hoc analysis presented in the paper is valuable from an explainable AI perspective. Evaluating whether BCI models capture clinically and biologically relevant signal characteristics is an important and emerging research direction, especially in the context of large foundation or generative models for neural signals.

- The analysis on source-level data, corresponding to specific and functionally relevant brain regions rather than only sensor/channel-level representations, provides stronger clinical relevance and interpretability.

## Weaknesses

- The authors apply channel attention before temporal attention, but the motivation behind this architectural choice is not clearly justified. It would be helpful to discuss how the results may vary if the ordering is reversed or if both are jointly modeled.

- The paper lacks clarity regarding its core contribution and intended application setting, making it difficult for readers to fully engage with the work and understand its broader significance.

- The analysis is largely qualitative. The authors should additionally report quantitative similarity metrics such as Dynamic Time Warping (DTW), Time Warp Edit Distance (TWED), or related sequence similarity measures to better evaluate the aggregate quality of the generated signals at the dataset level.

- While matching the distribution of handcrafted statistical or neurophysiological features is useful from an explainability perspective, the paper lacks downstream evaluation demonstrating whether the generated synthetic data is practically useful. For example, the authors could evaluate whether training with synthetic data improves downstream classification, decoding, or representation learning performance. Additionally, the utility and significance of the selected statistical/explainable features should be discussed more clearly.

- "Secondly, existing models are primarily evaluated on downstream task performance, which may not directly reflect the quality of learnt representations. High task performance does not necessarily imply that models capture intrinsic properties of neural signals, and task-specific optimisation during model design can introduce inductive biases that limit generalisation." The authors should support this claim with stronger references and a more balanced discussion, since large-scale foundation model training often enables the learning of broader inductive biases and transferable representations that generalize beyond specific training tasks and datasets.

- "This process is repeated to produce full-length sequences, which are then decoded via the tokeniser to obtain continuous MEG signals." The terminology is somewhat confusing because the framework appears closer to an autoencoder-style encoder-decoder setup rather than a tokenizer in the conventional language modeling sense. Using terminology more consistent with mainstream neural signal and representation learning literature would improve clarity.

- "For channel $c \in {1, \dots, C}$ and time index $t \in {1, \dots, L}$, embeddings are obtained via lookup tables." Using lookup tables for temporal embeddings differs from standard approaches in modern sequence modeling literature and may limit representational capacity because the temporal embeddings become arbitrarily fixed. While this design is reasonable for channel embeddings due to the limited number of channels, its suitability for temporal modeling is less clear. Furthermore, the embeddings are fused using summation, which may cause certain feature types to dominate others. The authors should justify this design choice more carefully, compare it with alternatives such as concatenation or projection-based fusion, and support the discussion with stronger intuition and references.

- The improvements reported over the baseline MEG-GPT are relatively small. Although the authors acknowledge this limitation, it remains difficult to conclusively establish meaningful performance gains based on the reported results.

- The authors keep several hyperparameters fixed without providing sufficient ablation studies or intuition behind the choices. In particular, the chosen sequence length of 16 appears potentially restrictive, as it provides the model with very limited temporal context. Since long-range temporal dependencies are important in neural signal modeling, the authors should investigate how varying the sequence length affects performance and representation quality.

## Suggestions

- The authors should formulate the overall idea and motivation more clearly, explicitly state the key contributions of the work, discuss the significance and practical relevance of the proposed approach, and provide stronger supporting references to justify several of the claims and arguments presented throughout the paper.

- The authors should conduct more rigorous experimentation across additional datasets and settings in order to better establish the robustness and generalizability of the proposed approach and the associated claims.

- The authors are encouraged to carefully address the weaknesses highlighted above, as doing so would substantially improve the clarity, credibility, and overall readability of the paper, making it easier for readers to engage with the work.

## References

1.) Lévy, Jarod, et al. "Brain-to-text decoding: A non-invasive approach via typing." arXiv preprint arXiv:2502.17480 (2025).
2.) Jiang, Wei-Bang, Liming Zhao, and Bao-Liang Lu. "Large brain model for learning generic representations with tremendous EEG data in BCI." International Conference on Learning Representations. Vol. 2024. 2024.
3.) Jiang, Wei-Bang, et al. "NeuroLM: A universal multi-task foundation model for bridging the gap between language and EEG signals." arXiv preprint arXiv:2409.00101 (2024).

## Justification for Score

The paper presents an overall interesting concept by proposing MEG-GPT2, an architecture for synthetic MEG data generation, and performs a useful post-hoc qualitative analysis demonstrating how the generated data resembles the original signals in terms of important neural characteristics and MEG-specific features. However, the paper currently lacks a clear narrative flow and sufficient intuition behind several architectural and methodological choices. Additionally, many claims are not adequately supported with relevant references or rigorous analysis. The evaluation is also relatively limited, which restricts the credibility, reliability, and generalizability of the conclusions presented in the paper.

---

### Official Review · Reviewer_Gwby · 2026-05-19
**A clear and meaningful work on Transformer-based MEG modeling.**

**Rating:** 8
**Confidence:** 4

**Review:**

This work proposes a decoder-only Transformer foundation model for MEG time series analysis. The results demonstrate that the model captures spectral and temporal dynamics of brain activity while encoding anatomically meaningful spatial structures and certain amplitude-based functional connectivity patterns. Overall, the paper is easy to follow, and the reviewer considers it a meaningful and well-executed contribution.

The review has several concerns:

1) Since MEG signals are inherently long sequences, the paper should provide a clearer explanation of how the signals are segmented and how the foundation model processes long-context temporal information.
2) It would be valuable to analyze and visualize the spectra of generated MEG signals across different brain regions. Such analysis could leverage existing neuroscientific knowledge to better validate the physiological plausibility and correctness of the generated signals.

---

### Official Review · Reviewer_8FhD · 2026-05-20
**Review of MEG-GPT-2**

**Rating:** 6
**Confidence:** 3

**Review:**

## Summary
This paper proposes MEG-GPT-2, a decoder-only foundation model for source-level MEG. The model uses discrete tokenization and factorized channel-time attention, and is trained on resting-state Cam-CAN MEG using next-token prediction. Instead of focusing on downstream tasks, the paper evaluates whether the model captures intrinsic neural structure through channel embeddings, attention maps, generated-signal PSDs, and connectivity-related analyses. The authors find that the model captures some spectral, temporal, and anatomical structure, but struggles with phase-sensitive inter-channel dependencies.

---
## Strengths
1. The paper is relevant to the workshop, and the topic is less studied compared with EEG.
2. The model design is reasonable and well-motivated in the paper. The factorized channel-time attention is appropriate for structured neural time series.
3. The paper’s focus on representation-level and generative evaluation is interesting. The authors correctly point out that downstream performance alone may not reveal whether a model captures neural signal properties such as spectral power, spatial organization, or connectivity.
4. The analysis of generated signals is useful, especially the comparison of real and generated PSDs and the discussion that phase-sensitive dependencies remain poorly captured.

---
## Areas for Improvement
1. The main limitation is the lack of evaluation of the model. The representation-level analyses are interesting, but without linear probing, fine-tuning, or transfer experiments, it is difficult to assess whether MEG-GPT-2 learns useful foundation-model representations. Moreover, the evidence for generalization is limited. The model is trained and evaluated on resting-state, eyes-closed Cam-CAN source-level MEG. This does not guarantee a transfer to task MEG, different datasets, different MEG systems, different source-reconstruction pipelines, sensor-space data, or clinical populations.
2. The generative evaluation is only partial. Matching PSDs and some amplitude-based connectivity statistics is visually useful, but it does not prove that generated signals are realistic in a broader neuroscientific sense. As for synthetic MEG, the paper lacks comparison with alternative reconstruction or generative baselines. It would be useful to compare against simpler signal-processing baselines, AR/VAR models, masked autoencoders, VAEs, diffusion models, or prior MEG generative models. Without such comparisons, it is hard to know whether MEG-GPT-2 itself is a strong generative model or mainly matches coarse statistics such as PSD.
3. The attention visualizations are hard to interpret. The paper shows channel embedding similarity and averaged channel-attention maps, but attention weights are not equivalent to functional connectivity or neural interaction. Since the attention matrices are averaged over batch, time, and heads, it is unclear what the underlying neuroscientific meaning remains.

---
## Detailed Comments
1. Please see the above comments.
2. Please clarify what can be concluded from this MEG FM trained only on resting-state Cam-CAN. The current setup does not show cross-task, cross-device, or cross-dataset transfer. As a foundation model, it is necessary to show its robust generalization, which is currently lacking in the paper.

---
## Justification of Score
Marginally above the acceptance threshold. The paper is relevant, clear, and explores generative foundation modeling for MEG. However, the current validation is limited. Without downstream tasks, cross-dataset/task evaluation, stronger generative baselines, or more rigorous interpretation of attention maps, it is difficult to judge whether the model is a useful MEG foundation model rather than a model that captures selected resting-state statistics.